# Long-Term Radiological Pulmonary Changes in Mechanically Ventilated Patients with Respiratory Failure due to SARS-CoV-2 Infection

**DOI:** 10.3390/biomedicines11102637

**Published:** 2023-09-26

**Authors:** Mircea Stoian, Adina Roman, Alina Boeriu, Danusia Onișor, Sergio Rareș Bandila, Dragoș Florin Babă, Iuliu Cocuz, Raluca Niculescu, Anamaria Costan, Sergiu Ștefan Laszlo, Dragoș Corău, Adina Stoian

**Affiliations:** 1Department of Anesthesiology and Intensive Care, George Emil Palade University of Medicine, Pharmacy, Sciences and Technology of Târgu Mureș, 540139 Targu Mures, Romania; mircea.stoian@umfst.ro; 2Gastroenterology Department, George Emil Palade University of Medicine, Pharmacy, Sciences and Technology of Târgu Mureș, 540142 Targu Mures, Romania; aboeriu@gmail.com (A.B.); danusia.onisor@umfst.ro (D.O.); 3Orthopedic Surgery and Traumatology Service, Marina Baixa Hospital, Av. Alcade En Jaume Botella Mayor, 03570 Villajoyosa, Spain; sergiob1976@gmail.com; 4Department of Cell and Molecular Biology, George Emil Palade University of Medicine, Pharmacy, Sciences and Technology of Târgu Mureș, 540142 Targu Mures, Romania; dragos-florin.baba@umfst.ro; 5Department of Pathophysiology, George Emil Palade University of Medicine, Pharmacy, Sciences and Technology of Târgu Mureș, 540136 Targu Mures, Romania; iuliu.cocuz@umfst.ro (I.C.); raluca.niculescu@umfst.ro (R.N.); adina.stoian@umfst.ro (A.S.); 6Faculty of Medicine, George Emil Palade University of Medicine, Pharmacy, Sciences and Technology of Târgu Mureș, 540142 Targu Mures, Romania; anamariacostan97@yahoo.com; 7Intensive Care Unit, Mureș County Hospital, Street Gheorghe Marinescu no 1, 540136 Targu Mures, Romania; dragos.corau@gmail.com

**Keywords:** severe acute respiratory syndrome coronavirus 2 (SARS-CoV-2), corona virus disease 2019 (COVID-19), acute respiratory distress syndrome (ARDS), intensive care unit (ICU), invasive ventilation, pulmonary fibrosis, respiratory failure

## Abstract

From the first reports of SARS-CoV-2, at the end of 2019 to the present, the global mortality associated with COVID-19 has reached 6,952,522 deaths as reported by the World Health Organization (WHO). Early intubation and mechanical ventilation can increase the survival rate of critically ill patients. This prospective study was carried out on 885 patients in the ICU of Mureș County Clinical Hospital, Romania. After applying inclusion and exclusion criteria, a total of 54 patients were included. Patients were monitored during hospitalization and at 6-month follow-up. We analyzed the relationship between invasive mechanical ventilation (IMV) and non-invasive mechanical ventilation (NIMV) and radiological changes on thoracic CT scans performed at 6-month follow-up and found no significant association. Regarding paraclinical analysis, there was a statistically significant association between patients grouped by IMV and ferritin level on day 1 of admission (*p* = 0.034), and between patients grouped by PaO_2_/FiO_2_ ratio with metabolic syndrome (*p* = 0.03) and the level of procalcitonin (*p* = 0.01). A significant proportion of patients with COVID-19 admitted to the ICU developed pulmonary fibrosis as observed at a 6-month evaluation. Patients with oxygen supplementation or mechanical ventilation require dynamic monitoring and radiological investigations, as there is a possibility of long-term pulmonary fibrosis that requires pharmacological interventions and finding new therapeutic alternatives.

## 1. Introduction

From the first reports of a novel coronavirus (severe acute respiratory syndrome coronavirus 2 [SARS-CoV-2]) at the end of 2019 to the present, 768,560,727 confirmed cases of infection have been recorded, and the global mortality associated with corona virus disease 2019 (COVID-19) has reached 6,952,522 deaths as reported by the World Health Organization (WHO) [1]. Even though most cases with mild symptoms recover in 1–2 weeks, some cases evolve unpredictably, with manifestations ranging from a lack of symptoms to complications in different organs; some cases even evolve towards severe respiratory failure requiring mechanical ventilation, frequently resulting in the death of the patient [2,3,4,5,6]. Like other coronaviruses that have caused severe outbreaks, SARS-CoV-2 primarily affects the respiratory system. COVID-19 was recognized early on based on its pulmonary manifestations, but the medical world quickly realized that it was a systemic disease that affected multiple organs, such as the nervous, cardiovascular, hematopoietic, gastrointestinal, and renal systems [7,8,9,10,11,12].

Histopathological and imaging abnormalities characteristic of lung fibrosis were found in survivors of the 2003 severe acute respiratory syndrome coronavirus 1 (SARS-CoV-1) outbreak [13]. Recent studies published after the SARS-CoV-2 pandemic also attest to the development of pulmonary fibrosis in patients who presented with lung damage or acute respiratory distress syndrome (ARDS) [11,14], suggesting that SARS viruses have greater potential to induce a lung fibrotic proliferative response than other respiratory viruses [15]. Sturgill et al. published a study in 2023 in which they found that the occurrence of fibrotic changes on chest CT imaging for COVID-19 survivors of pneumonia with ARDS is higher compared to patients with non-COVID ARDS etiologies. This fact is assumed to result from the pathophysiological aspects of the evolution of SARS-CoV2 infection to which the pharmaceutical treatment and the particularities of ventilation in these patients are added [16]. The most characteristic radiographic changes in pulmonary fibrosis include traction bronchiectasis, reticulations, honeycombing, and ground-glass opacities [15].

The alveoli–capillary membrane consists of the following structures:(1)The alveolar epithelium situated on its basement membrane.(2)The capillary endothelium that lays on its basement membrane.(3)A thin interstitium separating the two basement membranes, a connective tissue containing fibroblasts, macrophages, collagen, and elastic fibers [17].

Type I alveolar epithelial cells (AECIs) surround the alveolar air space and participate in gas exchange, while type II alveolar epithelial cells (AECIIs) are responsible for surfactant production and can differentiate into AECIs to repair the defects [18].

ARDS is characterized by bilateral pulmonary infiltrates with an acute onset and severe evolution, characterized by hypoxemia and non-cardiac pulmonary edema that require mechanical ventilation (sometimes prolonged) as supportive therapy [19,20]. The main stages of ARDS evolution are the following:(1)An exudative, early phase with disruption of the alveolocapillary membrane and edematous flooding of the alveolar spaces, followed by(2)A proliferative phase, with clearance of exudative fluid and an attempt to reestablish the alveolar barrier; then, in patients who survive,(3)A fibrotic phase characterized by fibroproliferation, with some patients undergoing excessive fibrotic process [20,21].

Early intubation and mechanical ventilation can increase the survival rate of critically ill patients with COVID-19 ARDS [22,23]. However, mechanical ventilation in the intensive care unit (ICU), a significant number of patients admitted in the ICU, and crowding in these units are associated with a higher mortality risk [2]. Different outcomes of mechanical ventilation in COVID-19 patients have been reported in publications worldwide [24]. In a study conducted in China, out of 344 patients hospitalized in the ICU, 100 (29.1%) required invasive mechanical ventilation [25]. Barotrauma associated with mechanical ventilation for respiratory failure due to COVID-19 was found in 40% of cases. These complications can occur even when protective mechanical ventilation recommendations are followed [26].

Data regarding long-term respiratory changes in mechanically ventilated patients with invasive or non-invasive positive pressure are limited [27]. Our study aimed to analyze and describe the main risk factors associated with the development of post-COVID-19 pulmonary fibrosis and gain a better understanding of its pathophysiology. A secondary objective was to analyze the relationship between mechanical ventilation and the degree of long-term pulmonary lesions in patients with a medium-to-severe form of COVID-19 treated in the ICU of Mureș Clinical County Hospital.

Is this subject still relevant? Unfortunately, the risk of new outbreaks of COVID-19 remains, even though the pandemic has been officially declared to have ended. Thus, we consider the publication of data about the incidence of pulmonary fibrosis in COVID-19 survivors who suffered from ARDS and the effect of mechanical ventilation on long-term respiratory function to be important, given the small number of articles published in the literature regarding this topic thus far.

## 2. Materials and Methods

The study protocol was approved by the Ethics Committee of the Mureș Clinical County Hospital. All procedures were conducted according to the principles of the Helsinki Declaration.

This prospective study was carried out in the ICU of Mureș County Clinical Hospital, Tîrgu Mureș, Romania, which contains 20 beds for critically ill patients. This ward functioned as an ICU for COVID-19 patients during the pandemic period. The study, lasting 1 year and 9 months, was conducted between April 2020 and December 2021. A total of 885 SARS-CoV-2-positive patients were admitted to the ICU. After reviewing the inclusion and exclusion criteria and eliminating the deceased patients, 91 were included in the study. Patients were monitored from the day of hospital discharge until 6 months after discharge. Six months after ICU discharge, the participants underwent a non-contrast thoracic CT scan. From this group, 35 patients were excluded because they did not attend the periodic evaluations and provide all the requested data or died.

Thus, the final study group included 56 adult patients (Figure 1) diagnosed with a medium/severe form of respiratory failure that fulfilled the criteria for ARDS according to the Berlin definition [19] and was caused by SARS-CoV-2 infection, with a positive reverse transcription polymerase-chain-reaction (RT-PCR) swab test, who were mechanically ventilated for more than 48 h during the hospitalization period (33 with non-invasive mechanical ventilation [NIMV] and 23 with invasive mechanical ventilation [IMV]). Pulmonary fibrosis was diagnosed according to radiological criteria by performing a computerized tomography (CT) scan 6 months after discharge and categorized according to severity.

Inclusion criteria: Adult patients over 18 years of age who gave their consent to enter the study, with a positive RT-PCR SARS-CoV-2 diagnostic test, moderate or severe respiratory failure due to SARS-CoV-2 infection, and invasive or non-invasive mechanical ventilation for more than 48 h during hospitalization and who underwent a radiological investigation at 6 months after discharge.

Exclusion criteria: Patients under 18 years of age, those who did not provide informed consent to enter the study, those who did not agree to the publication of their data, those who died during the first 6 months after discharge, those who did not attend periodic evaluations, and those with incomplete data.

### 2.1. Data Collection

Patient data collected during hospitalization included sex, age, urban or rural origin, existing comorbidities, Quick Sequential Organ Failure Assessment (qSOFA) score at admission [28], acute physiology and chronic health evaluation II (APACHE II) score completed in the first 24 h after admission in the ICU [29], the ratio of arterial oxygen partial pressure (PaO_2_) to fractional inspired oxygen (FiO_2_) (PaO_2_/FiO_2_), IMV/NIMV, radiological investigations (thoracic CT scan), and laboratory parameters (white blood cell [WBC] count, serum ferritin level, D-dimer level, fibrinogen, C-reactive protein [CRP], procalcitonin, lactate dehydrogenase [LDH], and serum creatinine level). Those who received interleukin (IL) 6 (IL-6) blockers were also registered.

Other collected data included the number of days spent in the ICU, the type of positive-pressure ventilation applied, and the number of hours spent on the ventilator during hospitalization.

The patients included in the study were evaluated 6 months after discharge, and the examination performed on this occasion comprised a radiological investigation (thoracic CT scan). Depending on the severity of the radiological changes, the severity of post-discharge pulmonary involvement was classified as follows:(1)Mild-to-moderate forms in those who presented pulmonary radiological changes of 10–50%.(2)Severe forms in patients presenting more than 50% pulmonary radiological changes.

After performing data collection, the patients who were administered mechanical ventilation were classified into two groups:(1)Those with NIMV(2)Those with IMV

### 2.2. Statistical Analysis

Software applications such as IBM SPSS Statistics v26 (New York, NY, USA) and Microsoft Office Excel 2019 (Washington, DC, USA) were used for statistical analysis. The statistical analyses consisted of an assessment of parametric variables (ANOVA test), describing the data as continuous (mean, standard deviation [SD], median, min/max) depending on their distribution. Quantitative variables were correlated using the Pearson correlation coefficient (rho), with alpha set at 0.05. A *p*-value ≤ 0.05 was considered significant. To assess the correlation between the distributions of the categorical variables, contingency tables, and the Chi-squared test were used.

## 3. Results

### 3.1. Participant Characteristics

The study was conducted on 56 adult patients (over 18 years of age) diagnosed with a medium-to-severe form of ARDS according to the Berlin definition caused by SARS-CoV-2, with a positive RT-PCR test, who were mechanically ventilated for more than 48 h. Thirty-three (58.9%) patients received NIMV and 23 (41.07%) received IMV.

The mean age of the included patients was 56.54 ± 12.83 years (a range of 24–81 years). The mean age of the patients with IMV was 58.70 ± 12.11 years, and those with NIMV were 55.03 ± 13.28 years, without significant differences between these two groups (*p* = 0.297). The gender distribution was 33 men (58.9%) and 23 women (41.1%). We did not find a significant association between the presence of pulmonary fibrosis and the patients’ gender. The age, sex, and environment (urban/rural) of the participants were not statistically significantly correlated with the APACHE II or qSOFA severity scores or the presence or absence of pulmonary fibrosis.

Regarding rural or urban origin, 16 patients (28.6%) came from rural areas, and 40 patients (71.4%) from urban areas. We did not find a significant association between the presence of pulmonary fibrosis and the patients’ origin (rural or urban) (*p* = 1.00).

### 3.2. Associated Pre-Existing Comorbidities

Out of the total 56 patients, 49 (87.50%) had associated conditions such as pre-existing pulmonary diseases (13; 23.21%), cardiovascular diseases (26; 46.43%), neurological diseases (4; 7.14%), kidney diseases (8; 14.29%), hematological diseases (12; 21.43%), rheumatological diseases (8; 14.29%), metabolic syndrome (31; 55.36%), and endocrinological diseases (5; 8.93%). Upon analyzing the correlations between the number of days spent in the ICU and the presence of comorbidities, we obtained the following results: an increased number of hospitalization days correlated with the presence of kidney diseases (*p* = 0.04) but not with the presence of neurological diseases (*p* = 1.00), pulmonary diseases (*p* = 0.10), hematological diseases (*p* = 0.68), rheumatological diseases (*p* = 0.64), metabolic syndrome (*p* = 0.51) or endocrinological diseases (*p* = 0.57).

The presence of kidney disease was correlated with the presence of mild/moderate pulmonary fibrosis (*p* = 0.04).

The patients’ characteristics and associated comorbidities are shown in Table 1 and Table 2.

### 3.3. The Duration of Mechanical Ventilation

Fifteen patients (26.79%) required less than 100 h of mechanical ventilation (IMV/NIMV), and 41 patients (73.21%) required more than 100 h of mechanical ventilation (IMV/NIMV). A total of 45 patients (80.36%) were hospitalized in the ICU for less than 10 days, and 11 (19.64%) patients were administered ventilation for more than 10 days.

The association between patients grouped by the mean days of hospitalization and the presence of pulmonary fibrosis was not statistically significant. IMV or NIMV were not correlated with the APACHE or SOPHA severity scores.

The associations between patients grouped by mean days of hospitalization and the number of hours of mechanical ventilation (*p* = 0.02, ANOVA) and the type of ventilation—NIMV (*p* = 0.01, ANOVA) and IMV (*p* = 0.01, ANOVA), were statistically significant.

Upon evaluating the association between the number of hours of mechanical ventilation and the occurrence of radiological changes suggestive of pulmonary fibrosis on the thoracic CT scan performed 6 months after discharge, we did not observe a statistically significant association between the number of hours of mechanical ventilation and the presence of mild pulmonary fibrosis (*p* = 1.00, ANOVA) or moderate/severe pulmonary fibrosis (*p* = 0.14, ANOVA).

### 3.4. Number of Days Spent in the ICU

A total of 45 patients (80.36%) were admitted to the ICU for less than 10 days, and 11 patients (19.64%) required a more extended hospitalization in the ICU (more than 10 days). There was no statistically significant association between patients grouped by the mean duration of hospitalization in the ICU and the presence of pulmonary fibrosis at 6 months, as evidenced by radiographic tests (*p* = 0.32, Chi-squared test). Although the degree of fibrosis was not significantly correlated, there was no association between patients grouped by the mean days of hospitalization and the level of SpO_2_ (*p* = 0.02, ANOVA test) and PaO_2_/FiO_2_ ratio of 100–200 (*p* = 0.01, ANOVA test).

Upon assessing the relationship between the patients grouped by the mean duration of hospitalization in the ICU and the laboratory values of some biochemical parameters (CRP, ferritin, fibrinogen, procalcitonin, D-dimers, LDH, glucose, aspartate aminotransferase [AST], alanine transaminase [ALT], blood urea nitrogen, and creatinine) determined at admission, there were no statistically significant correlations found.

### 3.5. PaO_2_/FiO_2_ Ratio

Regarding the PaO_2_/FiO_2_ ratio at the time of ICU admission, we found the following distribution of patients: 15 patients (26.79%) with PaO_2_/FiO_2_ < 100, 32 patients (57.14%) with PaO_2_/FiO_2_ between 100 and 200, eight patients (14.29%) with PaO_2_/FiO_2_ between 200 and 300, and no patients (0%) with PaO_2_/FiO_2_ >300.

Evaluating the PaO_2_/FiO_2_ ratio at admission to the ICU and the hours of mechanical ventilation showed no association between the two. Patients with a PaO_2_/FiO_2_ ratio at admission of below 100 mm Hg showed alterations in the radiological investigations performed 6 months after ICU discharge, but the association was not statistically significant.

The associations between the subgroup of patients grouped by PaO_2_/FiO_2_ ratio and the administration of IL-6 blocker treatment (*p* = 0.04, Chi-squared test) metabolic syndrome (*p* = 0.03, Chi-squared test) were statistically significant. Metabolic syndrome was defined as abdominal obesity, dyslipidemia, high blood pressure, and elevated fasting glucose.

Regarding paraclinical examination, the associations between the subgroup of patients grouped by PaO_2_/FiO_2_ ratio and the serum level of urea at admission (*p* = 0.007, Chi-squared test), the level of procalcitonin (*p* = 0.01, Chi-squared test), and the levels of AST and ALT at admission (*p* = 0.002 and 0.02, respectively, Chi-squared test), were statistically significant.

The association between the subgroup of patients grouped by PaO_2_/FiO_2_ ratio and the presence of pulmonary fibrosis, regardless of severity, was not statistically significant.

### 3.6. Radiological Evaluation of Patients 6 Months after Discharge

All the patients underwent a thoracic CT scan at 6 months after discharge. Of these patients, seven (12.50%) presented with no fibrosis, 38 (67.86%) presented with mild/moderate fibrotic changes, and 11 (19.64%) presented with severe pulmonary fibrosis. Furthermore, 87.5% of the patients treated with IMV or NIMV in the ICU presented fibrotic changes in the radiological investigations performed 6 months after ICU discharge.

All CT scans performed at admission and 6-month follow-up were performed on CT Siemens Somatom go. Top 128 slice provides detailed imaging of the thoracic examinations performed in Mures County Clinical Hospital.

The top image describes in a native CT scan, coronal section, pulmonary window—the presence of ground-glass opacity (as pointed by arrows in Figure 2—top), parenchymal and subpleural bands, and reticular abnormality (Figure 2—top).

CT at 3-month follow-up after hospitalization, native CT scan, coronal section, pulmonary window—shows a patchy nodular consolidation (as pointed by arrows in Figure 2—bottom), moderate residual ground-glass opacity, and a honeycombing sign with architectural distortion (Figure 2—bottom).

### 3.7. Evaluation of the Patients with NIMV

We analyzed the relationship between NIMV and radiological changes on the thoracic CT scan performed 6 months after discharge from the ICU and found no significant association. The association between the subgroup of patients grouped by NIMV and the ferritin level at admission was statistically significant (*p* = 0.03, Chi-squared test), but there were no statistically significant correlations between the degree of fibrosis and other laboratory tests. There were no statistically significant differences between NIMV and IMV regarding pulmonary fibrosis or the number of hours of mechanical ventilation.

### 3.8. Evaluation of the Patients with IMV

We analyzed the relationship between IMV and radiological changes on the thoracic CT scan performed 6 months after discharge from the ICU and found no significant association. IMV was also not correlated with pulmonary fibrosis or the status of O_2_-dependent participants at discharge. The association between patients grouped by IMV and the ferritin level on day 1 of admission (*p* = 0.034, Chi-squared test) was statistically significant, but not with ferritin on day 3 of admission (*p* = 0.44, Chi-squared test), as seen in Table 3.

### 3.9. Paraclinical Laboratory Examinations

Regarding the studied paraclinical parameters, we established a statistically significant association between patients grouped by CRP level and LDH at admission (*p* = 0.03, Chi-squared test), using cut-off values as seen in Table 3.

The association between patients grouped by ferritin levels on day 1 of admission and ferritin levels on day 3 (*p* = 0.005, Chi-squared test) was statistically significant.

The association between patients grouped by procalcitonin levels and creatinine levels at admission (*p* = 0.04, Chi-squared test) was statistically significant.

Regarding liver panel tests, we established a statistically significant association between ALT levels on both day 1 and day 3 of admission and AST levels on day 1 and day 3 of admission (*p* = 0.01, Chi-squared test).

There was no statistically significant association between patients grouped by leucocyte count and the presence of pulmonary fibrosis, the number of hours of mechanical ventilation, or the type of ventilation used.

There was no statistically significant association between the patients grouped by the status of O_2_-dependent participants at discharge and the degree of pulmonary fibrosis (mild/moderate or severe). The evolution of different paraclinical examinations performed on the patients is presented in Table 3.

## 4. Discussion

### 4.1. Risk Factors for Pulmonary Fibrosis

The lungs of COVID-19 patients are damaged due to the viral infection that evolves severely with cytokine storm and the overlapping of mechanical ventilation and bacterial coinfections. Several other factors contribute to pulmonary fibrosis in patients admitted to the ICU, including post-thrombotic events with microvascular damage and alveoli–capillary membrane dysfunction, post-ischemic effects, corticosteroid use, and prolonged ICU stay [11,14]. All these factors act simultaneously, determining the progression of the injury towards fibroproliferation.

#### 4.1.1. ARDS

ARDS was present in all patients admitted to ICU (100%). ARDS is one of the most important predictors of mortality in COVID-19 patients [21]. AECIIs express more abundant angiotensin-converting enzyme 2 (ACE2) receptors, resulting in the higher infectivity of these cells, which may explain the defective surfactant production and re-epithelization of the broncho-alveolar epithelium after injury [30,31,32,33]. Epithelial injury is accompanied by endothelial injury, microthrombosis, subsequent pulmonary edema, and vascular leak [21]. ARDS is reported to occur in up to 14% of COVID-19 cases and is a potentially fatal condition [34].

A hallmark of COVID-19 is extensive alveolar epithelial cell injury with secondary fibrotic proliferation, indicating the potential for chronic alveolar and vascular remodeling leading to pulmonary fibrosis and /or pulmonary hypertension [35,36,37]. According to the literature, pulmonary fibrosis can develop immediately after discharge or in the subsequent weeks, but in some cases, it improves in the months following COVID-19 recovery [11].

Studies estimate that 70–80% of the survivors of severe COVID-19 continue to suffer from long-term postinfectious complications [3,38], of which pulmonary fibrosis is among the most severe and frequently reported [11,39]. The exact prevalence of this sequela is not yet fully established, but some studies reported a prevalence of 62% after SARS-CoV-2 infection [40].

#### 4.1.2. Hypercoagulability

D-dimers were increased in 29 out of 56 patients on the first day of ICU admission, and the number increased to 33 out of 56 on the third day. COVID-19 induces a prothrombotic state, leading to micro- and macrovascular thrombosis, including pulmonary artery thrombi and fibrin microthrombi in small pulmonary arteries [41]. Epidemiological studies have shown a link between thrombotic vascular events and idiopathic pulmonary fibrosis (IPF), with a central role played by the coagulation and fibrinolysis systems in the wound healing and repair processes. A correlation was observed between IPF and the prothrombotic state, but trials with anticoagulation treatment regarding both the progression of fibrotic and thrombotic risks are inconsistent [42]. The logical question that arises in the context of coagulopathies and the hypercoagulable status with inflammatory thrombosis in the pulmonary vasculature associated with COVID-19 is whether this also contributes to the occurrence of pulmonary fibrosis.

#### 4.1.3. Role of Oxygen

All our patients required oxygen supplementation during hospitalization. Moreover, 50% of them presented with oxygen dependency at discharge. The impairment of gas exchange in ARDS culminates in significant hypoxemia. Hypoxemia secondary to lung disease activates hypoxia-inducible factor 1-alpha (HIF-1-α), stimulating tissue fibrosis and activating fibroblasts and ECM deposition [41]. In IPF, the dysregulated expression of HIF-1-α augments myofibroblast differentiation [43]. On the other hand, it was proven that supplemental oxygen used to treat newborns with respiratory disorders and exposure to high-concentration oxygen are associated with inflammation, acute lung injury, and lung fibrosis [44]. The questions that arise in COVID-19 are “How much oxygen is required?” and “How best to titrate it to be useful?” Prolonged oxygen therapy can also increase hyperoxia-induced ROS levels, leading to protein denaturation and the breaking down of nucleic acids [45]. Typically, in viral respiratory diseases, ROS enhance the phagocytosis and activity of immune cells, but excessive oxygenation can cause an imbalance, with extra amounts of ROS that alter redox homeostasis and contribute to alveolocapillary membrane destruction [46]. This combination of SARS-CoV-2 infection and excess ROS production exacerbates cell apoptosis of the alveolar epithelium [47]. An overcorrection of hypoxemia can produce oxygen-induced ARDS and VILI [48].

#### 4.1.4. Age and Sex

Some studies associate the occurrence of pulmonary fibrosis with advanced age, an underlying history of diabetes mellitus, and respiratory and cardiovascular diseases [14,49]. In our study, the mean age of fibrotic patients was 57.08 ± 12.63 years, and that of non-fibrotic patients was 52.71 ± 14.60 years. These values are similar to the ages reported by Amin et al. (59 years in those with fibrotic changes compared with 48.5 years in those without fibrotic changes) [11]. Increased age is associated with lung parenchymal stiffening and the profibrotic potential of lung fibroblasts and is considered a major risk factor for the occurrence and progression of pulmonary fibrosis [14,50]. Experimental studies using mouse models reported that fibroblasts and myofibroblasts were more resistant to apoptosis in older animals [51]. Cilli et al. showed that older people with pre-existing chronic respiratory diseases such as IPF represent a vulnerable population group for COVID-19 with a more severe course of the disease and increased mortality rates [52].

The data regarding the association between sex and pulmonary fibrosis after COVID-19 are contradictory, with some studies suggesting a higher prevalence of pulmonary fibrosis in men, while others do not highlight differences between the sexes in favor of men [11,53]. Other studies also correlate the occurrence of pulmonary fibrosis with risk factors such as older age, mechanical ventilation, and the female sex instead of the male sex [54,55]. In our study, no higher prevalence of pulmonary fibrosis in either female or male patients was observed; thus, we could not establish sex or even age as a risk factor for developing pulmonary fibrosis. The mean age of patients included in the study who developed pulmonary fibrosis is similar to that reported in other studies where age proved to be a risk factor for the occurrence of pulmonary fibrosis, and the percentage of men (58.9%) included in the study was higher than that of women.

#### 4.1.5. Associated Comorbidities

Amin et al. reported that of all the comorbidities tested, only chronic obstructive pulmonary disease (COPD) was linked to an increased risk of pulmonary fibrosis [11]. Other studies reported associations between hypertension and pulmonary fibrosis [56].

In our study, mild/moderate pulmonary fibrosis was correlated with renal disease, whereas cardiovascular and pulmonary comorbidities were not correlated with the presence of pulmonary fibrosis. Patients with renal disease also presented with a greater length of hospitalization. We found a correlation between chronic kidney disease and the presence of mild/moderate pulmonary fibrosis, and the studies also suggest that it is correlated with a poorer outcome of COVID-19 [57].

#### 4.1.6. Treatment and Ventilatory Support

Previous reports emphasized that the severity of COVID-19 correlates with the risk of pulmonary fibrosis, which is also evident from our study, with all the included subjects being critically ill patients [58]. According to Zhou et al., 80% of patients infected with SARS-CoV-2 present with mild forms of the disease, 14% develop more severe symptoms, and 6% develop critical forms [59]. ICU admission is necessary for approximately 5–12%, depending on the criteria of the adopted local protocols, with most such patients being likely to benefit from ventilatory support [60,61]. Other risk factors mentioned in previous studies include ICU admission, supplemental oxygen requirement, NIMV or IMV application, and hospitalization length [53,62,63]. The patients included in the present study were critically ill, required supplemental oxygen, and benefited from NIMV or IMV during hospitalization in the ICU.

All our patients admitted to the ICU benefitted from dexamethasone therapy in accordance with the national protocol valid during the outbreak, and 19.6% received IL-6 pharmacological inhibitors. There were no statistically significant differences between patients receiving either therapeutic scheme. However, there was a correlation between a PaO_2_/FiO_2_ ratio of 100–200 and IL-6 receptor blockade treatment, as this treatment was initially recommended only for patients with severe forms of disease. The results of studies that evaluated the administration of corticosteroids and selective IL-6 inhibitors are few and conflicting [14]. Based on their mechanism of action, immunomodulatory agents are assumed to reduce inflammation and thus the duration of mechanical ventilation; however, it is not clear if the use of corticosteroids in the stage of ARDS reduces the risk of pulmonary fibrosis and whether the simultaneous use of corticosteroids and antifibrotics might influence long-term outcomes is currently being investigated [14]. On the contrary, in a recent meta-analysis, steroid therapy increased the risk of pulmonary fibrosis threefold [11]. McGroder et al. found that lung fibrosis was correlated with the male sex, higher SOFA scores at admission, steroid treatment, and anti-IL6 receptor blockade [15].

In 2021, Torres Castro et al. published a meta-analysis showing that 15% of COVID-19 survivors had a restrictive ventilation pattern, while 7% presented an obstructive pattern of ventilation [64]. The alteration of pulmonary functional tests can also suggest the involvement of the cardiovascular system and neuromuscular dysfunction, which can be secondary to the neurological sequelae that appear in critically ill patients [65,66].

All our patients received IMV/NIMV. Moreover, 73.21% of patients required more than 100 h of mechanical ventilation.

Duration of mechanical ventilation of over 5 days in patients with ARDS correlated with the presence of pulmonary fibrosis in 53% of ventilated patients, according to Papazian et al. [67], and the presence of fibrosis led to an increase in mortality of up to 57% [68]. It seems that, at least partially, the development of pulmonary fibrosis in ARDS can be attributed to mechanical ventilation and not only to inflammation and secondary repair processes [20,69]. The underlying pathophysiology is multifactorial, and mechanical ventilation seems to contribute to the supplementary trauma of the lungs [41], an entity known as mechanical ventilation-induced lung injury (VILI) [20,70].

The forces generated through mechanical ventilation with high transpulmonary pressure produce injury and alveolar overdistension with barotrauma, influencing the course of lung remodeling [71]. The tight junctions at the alveolar level are distorted, and the epithelial layer is interrupted and damaged, which leads to the remodeling and reorganization processes of the extracellular matrix (ECM) proteins [20,72,73]. The pulmonary ECM consists of fibronectin, elastin, collagen fibers, proteoglycans, and laminin [74], and these elements are activated by mechanical ventilation and interact with growth factors (like transforming growth factor-beta [TGF-β]), resulting in a remodeling process [75,76].

An increase in procollagen type III (PCIII) is an early marker of the fibrosis process [20,77]. Mechanical stress in isolated rat lungs was associated in experimental studies with enhanced PCIII gene expression [74,77,78]. Other studies in open-chest rabbits showed that mechanical ventilation using a high positive end-expiratory pressure is also associated with increased PCIII, procollagen type IV (PCIV) (a fibroblast growth factor), and transforming growth factor β1 (TGF-β1) gene expression [79]. In contrast, ventilation with lower pressures did not influence the expression of these genes [80].

Although we could not demonstrate the presence of VILI, 12.5% of patients in our study had no fibrosis, 67.86% presented with a minor-to-moderate degree of fibrosis, and 19.64% presented with severe fibrosis at follow-up examination.

The length of mechanical ventilation was not correlated in our group of patients with oxygen dependency at discharge or the comorbidities of the patients. However, the increased number of patients who developed pulmonary fibrosis is an additional argument for the multifactorial etiology of pulmonary fibrosis, including VILI.

#### 4.1.7. Disease Severity

A study published in April 2021 demonstrated that the severity of the initial illness as quantified by the SOFA score, the level of LDL at admission, and the duration of mechanical ventilation are correlated with a higher risk of fibrotic-like radiographic abnormalities at 4 months after COVID-19 [15]. Moreover, the incidence of pulmonary fibrosis is 72% in those who underwent mechanical ventilation. The predicted risk of fibrotic changes varied linearly with the duration of mechanical ventilation through the first 20 days and then presented a plateau for more extended periods of mechanical ventilation [15]. In another study published in 2020 on patients admitted to an ICU in Lombardy, Italy, 88.4% (1150 out of 1300) required mechanical ventilation [81]. Out of these patients, 27 ventilated for ARDS were followed up at 110–267 days after extubation, and 23 (85%) presented pulmonary fibrosis [82], which strongly aligns with our data. Only a few studies have monitored the evolution of ARDS developed secondary to SARS-CoV-2 infection over several months in mechanically ventilated patients, but even these scarce data are helpful to provide us with a comparison term. In addition, our study reveals new information regarding the incidence of fibrosis in such patients.

### 4.2. Follow Up

Our study evaluated the patients 6 months after hospitalization and found an increased percentage of pulmonary fibrosis. The best time to identify irreversible pulmonary fibrosis is not well established, with some experts suggesting pulmonary functional tests and radiological investigations at 3, 6, and 12 months after an acute infectious episode [14]. Patients with pulmonary fibrosis presented lower respiratory volumes (forced vital capacity [FVC]) on pulmonary functional tests and diminished total lung capacity (TLC) [14].

The prevalence of pulmonary fibrosis in our study was 87.5%, of which 19.64% presented with severe pulmonary fibrosis. In a meta-analysis of published studies regarding pulmonary fibrosis and SARS-CoV-2 infection performed by Amin et al. in 2022, the overall prevalence of pulmonary fibrosis was found to be 44.9% [11]. Most survivors of COVID-19 suffer from postinfectious acute complications or long-term effects, especially those who present severe forms of the disease with ICU admission [14,65,83]. The most critical consequences of pulmonary fibrosis are the presence of dyspnea, limitations in physical performance (with reduced effort capacity as evidenced on pulmonary respiratory functional tests), and long-term alterations of the postinfectious quality of life. However, the good news is that in most patients, pulmonary dysfunction gradually improves over months [12]. Nabahati et al. reported that pulmonary fibrosis diminished at 6-month follow-up in 33.9% of the patients compared with those at 3-month follow-up as per thoracic CT scans [84], but patients whose fibrosis worsens over time have also been identified [11]. In 2020, Fang et al. published a study that followed 12 patients diagnosed with a severe form of COVID-19 (six treated with NIMV, four treated with IMV, and one treated with extracorporeal membrane oxygenation) and found proof of pulmonary fibrosis on CT scans performed 2 months after infection in all patients [85]. These results are also consistent with our patients’ high rate of pulmonary fibrosis.

### 4.3. Preliminary Findings

In our study, all included patients were critically ill patients diagnosed with severe forms of ARDS, and all needed ventilatory support. Supportive oxygen therapy was used in all ventilated patients. The SOFA and APACHE scores of the monitored patients attested to the severity of the disease (as illustrated in Table 1). The treatment of these patients was challenging; critical patients had to be dealt with, among whom the mortality rate was very high, and those who survived suffered many injuries.

Fibrotic pulmonary changes seem to be a characteristic of SARS-CoV-2 infection, being more frequent in this case than in other viral cases of pneumonia (they are reported in 20% of patients after avian influenza A (H7N9) infection and 8% of patients following SARS infection) [86,87].

Pulmonary fibrosis may be a part of the long-COVID syndrome, wherein patients experience persistent symptoms and long-term consequences [3]. Patients with long-term pulmonary involvement can place an enormous burden on healthcare systems and require long-term specialty care.

### 4.4. General Considerations and Pathophysiological Aspects of Pulmonary Fibrosis

#### 4.4.1. Role of Immune Cells and Cytokines

SARS-CoV-2 binds to human cells through ACE2 receptors found in many tissues such as brain, heart, lung, intestine, and kidney [12,83]. It enters through the ACE2 receptors in the pulmonary epithelium, leading to respiratory tract infection [3,14,83]. The main targets of SARS-CoV-2 in the respiratory airways are AECIIs and alveolar macrophages [88]. The virus is processed by antigen-presenting cells (APCs) and recognized by toll-like receptor 7 (TLR7) from the macrophages, enhancing intracellular signaling and activating nuclear-factor kappa B (NF-KB), with increased expression of proinflammatory cytokines [12,89,90]. APCs interact with the CD4+ cell subset of T-cells through the intermedium of major histocompatibility complex II (MHCII). CD4+ cells further proliferate and differentiate into the T helper 1 (Th1), Th2, and Th17 subclasses [91]. Th1 is responsible for the increased production of tumor necrosis factor (TNF), interferon-gamma (IFN-γ), and IL-2, while Th2 stimulates the production of IL-4, IL-5, IL-10, and IL-13, with B-cell activation and antibody production [12,92,93,94].

Classically, macrophages are activated via two pathways:(1)M1 (classical)(2)M2 (alternative)

Differentiation of the M1 phenotype is stimulated by TLR ligands and IFN-γ and is characterized by the production of proinflammatory cytokines, reactive oxygen species (ROS), reactive nitrogen, and Th1 stimulation [88]. The M1 phenotype is involved in initiating and maintaining inflammation [95]. Differentiation of the M2 phenotype is stimulated by IL-4 and IL-13 and is characterized by phagocytic activity, with a role in the inflammation resolution process [95].

Pulmonary macrophages recruited and stimulated by Th2 cytokines shift to an M2 phenotype, which also has a role in pulmonary remodeling and fibrosis in ARDS patients [20,96]. In the subsequent stages, the virus infiltrates alveolar cells (especially type 2 alveolar AECIIs), producing diffuse alveolar damage (DAD) [97], and inflammatory cytokines combine their activity with virus-induced lesions to produce lung damage [12]. The alveolar epithelial damage with the loss of epithelial barrier function produced by the virus is followed by the activation of alveolar macrophages, which initiate the process of phagocytosis of alveolar debris resulting from virus injury and release cytokines (especially IL-1 and TNF-α) and growth factors that stimulate the native cells and the regeneration of connective tissue to repair the defects [12,98,99]. The cytokine storm is the result of an immune hyperreaction due to the uncontrolled release of high amounts of cytokines that leads to multiorgan dysfunction [100], with the degree of elevation of proinflammatory cytokines correlating with the degree of disease severity [12,101,102,103]. The release of IL-1 and TNF-α induces the activation of adhesion molecules such as vascular adhesion molecule 1 (VCAM-1), intercellular adhesion molecule 1 (ICAM-1), and selectin, which mediate leukocyte recruitment and marginalization, rolling, and extravasation in the alveolar spaces [12]. The lesions of the alveolocapillary membrane increase the permeability of the capillary endothelium, which, in conjunction with histamine, leukotrienes, and bradykinin, allow the leakage of fluid in the interstitial and lung alveolar spaces [104]. Neutrophil infiltration in the lungs during ARDS could also modulate the fibrotic process [105,106].

#### 4.4.2. Pulmonary Fibrosis Mechanism

Fibrosis is the consequence of an aberrant wound-healing process that follows a lung injury characterized by a distortion of normal lung architecture with lung dysfunction [12]. Fibrotic changes are secondary to inflammation, with epithelial and endothelial injury and damage to the alveolo-capillary membrane [20]. Post-COVID-19 pulmonary fibrosis is characterized by persistent fibrotic radiological/tomographic changes with functional impairment in respiratory tests at follow-up [14].

Essential components of the ECM of the basement membrane of the lungs (laminins, collagen VI, and fibronectin) are downregulated, while others are upregulated (MMP2, MMP8, and cathepsin proteins) [107]. Proinflammatory cytokines in SARS-CoV-2 infection are also believed to increase matrix metallopeptidase 1 (MMP1) and MMP7 expression, which degrades the ECM and contributes to airway remodeling and pulmonary fibrosis [14,108]. Increased CRP, IL-6, and LDH levels in the presence of systemic inflammation may activate fibroblast proliferation during the healing process of lung injury [14,109]. The proliferation of fibroblastic tissue with the excessive deposition of ECM leads to interlobular septal thickening, traction bronchiectasis with the modification of typical lung architecture, and lung fibrosis [14,110]. TGF-β is hypothesized to be directly amplified by the SARS-CoV-2 nucleocapsid protein and is also upregulated by angiotensin II, which increases in the lungs in a compensatory manner due to ACE2 receptor downregulation caused by the virus [111]. TGF-β is also produced by activated alveolar macrophages and bronchial epithelial cells and acts as a significant profibrotic stimulus [14] by stimulating the proliferation and migration of fibroblasts and activation of myofibroblasts and regulating collagen, fibronectin, and elastin formation and deposition. It also stimulates excessive ECM formation and deposition and prevents its degradation by matrix metalloproteinases, with all these factors leading to an excessive accumulation of fibrotic scar tissue [35,99,112].

Lung fibroproliferation was associated with ventilator dependency and increased mortality [113]. The pressures used in mechanical ventilation with cellular damage favor the release of inflammatory cytokines [114], accentuating lung injury and leading to inflammatory cell recruitment. Activating type 1 and type 2 helper T-cells (Th1 and Th2) causes the release of additional chemokines and growth factors, producing additional biotrauma associated with barotrauma [20,114]. ROS are produced by activated immune cells in the context of inflammation and induce alveolar epithelial apoptosis with neutrophil degranulation and oxidative stress, promoting the secretion of profibrotic cytokines with fibroblast activation [115].

Fibroblasts are mesenchymal cells found in different tissues, including the lungs. They are essential in tissue repair, secreting, and regulating the ECM [116]. Residual lung fibroblasts are activated and proliferate in response to growth factors (fibroblast growth factor [FGF], TGF-β, platelet-derived growth factor [PDGF], and epidermal growth factor [EGF]) and Th2 cytokines such as IL-1 [117] and differentiate into myofibroblasts [118]. Circulating fibrocytes originating in the bone marrow can travel from the peripheral blood to the lung and transform into interstitial fibroblasts [119,120]. The presence of fibrocytes in the broncho-alveolar lavage fluid of mechanically ventilated ARDS patients is an independent predictor of mortality [121]. Once activated, fibroblasts synthesize collagen, fibronectin, ECM components, and mediators of the repair process, such as growth factors [110]. The development of pulmonary fibrosis was correlated with increased levels of TGF-β and PCIII [20]. The growth factors vascular endothelial growth factor (VEGF) and FGF stimulate the proliferation of intact endothelial cells and the process of pulmonary capillary angiogenesis [122], while TGF-α and epidermal growth factor (EGF) stimulate the bronchiolar stem cells to regenerate the injured alveolar epithelium [123].

In our study, as observed on the CT scans, most patients presented with a minor-to-moderate degree of pulmonary fibrosis, and only 12.5% presented no fibrosis. The degree of fibrosis was not correlated with the patient’s comorbidities, paraclinical examinations, or the number of hours of mechanical ventilation required.

### 4.5. Therapeutic Perspective

Kooistra et al. performed a study that revealed the upregulation of coagulation, inflammatory, and neutrophil extracellular trap-related pathways in patients with pulmonary fibrosis associated with severe COVID-19. They also proved that early dexamethasone treatment did not influence the severity or the incidence of pulmonary fibrosis. Instead, prednisone treatment initiated in patients with early suspicion of pulmonary fibrosis (both from the group on dexamethasone treatment and from the group that did not receive dexamethasone) was correlated with the reduction of fibrosis biomarkers and specific genes (matrix metalloproteinase 8 [MMP8], phosphodiesterase 4D [PDE4D], cysteine-rich secretory protein [CRISP3], B-cell lymphoma 2 like protein 15 [BCL2L15] that were previously upregulated towards those observed in non-pulmonary fibrosis group [124]. Other studies to prevent pulmonary fibrosis suggest that inhibitors of upregulated genes (MMP8, PDE4) in patients with pulmonary fibrosis might be a therapeutic target [124,125,126].

Another recent study showed that epidermic growth factor receptor (EGFR) positivity and pulmonary fibrosis are associated with increased D-dimer levels, CRP levels, and prolonged ICU stay. EGF has essential roles in cell differentiation, division, and migration and acts through its receptors (EGFR), stimulating fibroblast and vascular endothelial cell growth and proliferation. These findings can provide preliminary data for developing and using antifibrotic drugs in COVID-19 ARDS patients [127].

## 5. Limitations of the Study

Because critically ill patients have a low survival rate, the study included only a limited number of patients, leading to a small sample size. Additionally, it was conducted in a single medical unit and was not a multicenter study. There was also no clear distinction between pre-existing symptoms and past comorbidities with baseline testing and those caused by COVID-19 (a baseline status would be required to eliminate bias).

The evaluation of the patients was performed at 6 months; however, these patients would generally require a more extended follow-up period. Although the 56 patients regained a close-to-normal quality of life, they still presented symptoms like dyspnea and fatigue, and whether their lung lesions will disappear or persist over time requires additional studies that include such patients. More comprehensive studies or meta-analyses, including patients who underwent mechanical ventilation, are required to understand the process better.

## 6. Conclusions

A significant proportion of patients with COVID-19 admitted to the ICU developed pulmonary fibrosis, as observed at a 6-month post-discharge evaluation. The most important risk factors for pulmonary fibrosis identified in the medical literature were also characteristic of our patients and included severe forms of COVID-19, ARDS, oxygen supplementation, and mechanical ventilation. Such patients require dynamic monitoring, as there is a possibility of long-term pulmonary fibrosis that requires pharmacological interventions and finding new therapeutic alternatives to prevent and reduce the incidence of this complication. Radiological investigations and pulmonary functional tests should be performed to evaluate and diagnose pulmonary fibrosis in cases of SARS-CoV-2 infection, especially in severe forms.

## Figures and Tables

**Figure 1 biomedicines-11-02637-f001:**
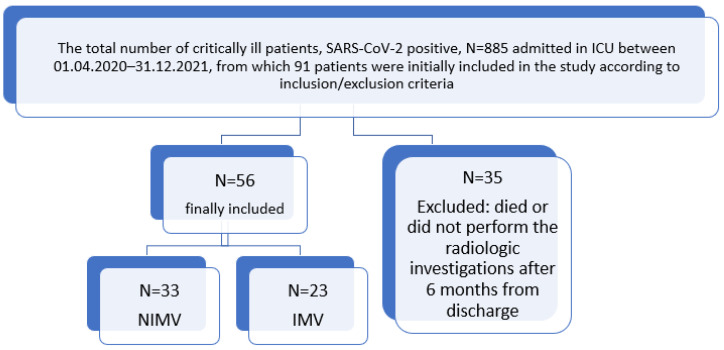
Flow diagram of study participants.

**Figure 2 biomedicines-11-02637-f002:**
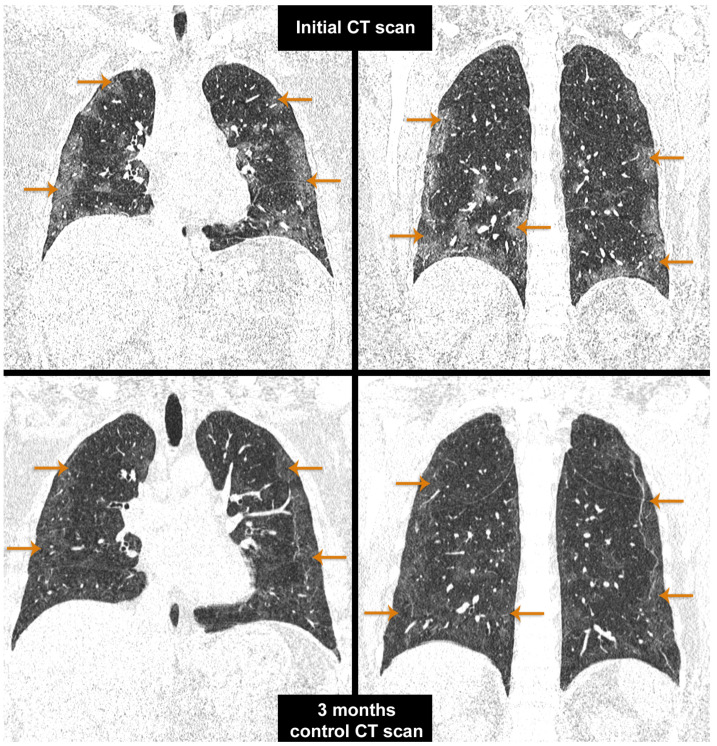
CT scan—initial vs. control (3-month follow-up after hospitalization).

**Table 1 biomedicines-11-02637-t001:** Patients’ age and severity score.

Variable (*n* = 56)	Mean ± Standard Deviation (Min–Max)
Age (years)	56.54 ± 12.83 (24–81)
APACHE II	22.41 ± 9.48 (10–45)
q-SOFA	1.96 ± 0.73 (1–3)

**Table 2 biomedicines-11-02637-t002:** Patients’ clinical characteristics and associated comorbidities.

Variable (*n* = 56)	Count (Number of Patients)	Percentage (%)
Sex (Male:Female)	33:23	58.93%:41.07%
ICU admission days		
<10 days	45	80.36%
≥10 days	11	19.64%
Patients with mechanical ventilation		
<100 h	15	26.79%
≥100 h	41	73.21%
Patients with NIMV vs. IMV		
NIMV:IMV	33:23	58.93%:41.07%
Oxygen dependency at discharge	28	50%
Pulmonary fibrosis		
No fibrosis	7	12.50%
Minor–Moderate	38	67.86%
Severe	11	19.64%
PaO_2_/FiO_2_		
<100	15	26.79%
100–200	32	57.14%
200–300	8	14.29%
Comorbidities		
Cardiovascular	26	46.43%
Pulmonary	13	23.21%
Neurological	4	7.14%
Renal	8	14.29%
Hematological	12	21.43%
Rheumatological	8	14.29%
Metabolic syndrome	31	55.36%
Endocrine	5	8.93%
Other comorbidities	26	46.43%

**Table 3 biomedicines-11-02637-t003:** Paraclinical examinations performed on day 1 and day 3 of admission.

	Cut-Off Value	Day 1 (Number of Patients)	Day 3 (Number of Patients)	Evolution
CRP (mg/dL)	<0.5	7	7	↔
	≥0.5	49	49	↔
Fibrinogen (mg/dL)	<196	3	3	↔
	196–372	14	16	↗
	>372	39	37	↘
Ferritin (ng/mL)	<300	7	8	↗
	≥300	49	48	↘
Procalcitonin (ng/mL)	<0.5	31	32	↗
	≥0.5	25	24	↘
D-Dimers	Negative	27	23	↘
	Positive	29	33	↗
LDH (U/L)	<220	9	12	↗
	≥220	47	44	↘
Leucocytes (/mm^3^)	<4000	3	2	↘
	4000–10000	11	17	↗
	>10000	42	37	↘
Lymphocytes (%)	<25%	47	46	↘
	25–45%	4	9	↗
	>45%	5	1	↗
Neutrophils (%)	<40%	3	1	↗
	40–75%	15	10	↘
	>75%	38	45	↗
AST (U/L)	<34	19	20	↗
	≥34	37	36	↘
ALT (U/L)	<55	27	27	↔
	≥55	29	29	↔
Creatinine (mg/dL)	<1	38	45	↗
	≥1	18	11	↘
Urea (mg/dL)	<55	29	23	↘
	≥55	27	33	↗
Glucose (mg/dL)	<80	0	1	↗
	80–110	15	11	↘
	>110	41	44	↗

↔ = stationary; ↗ = increased; ↘ = decreased.

## Data Availability

No new data were created or analyzed in this study. Data sharing is not applicable to this article.

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
