# Peer review of "Long-Term Radiological Pulmonary Changes in Mechanically Ventilated Patients with Respiratory Failure due to SARS-CoV-2 Infection"

_biomedicines, 2023, doi:10.3390/biomedicines11102637_

Round 1

Reviewer 1 Report

Overall interesting study worthy of publications, although a series of issues require some work. 

A major problem is that the statically treatments are somewhat hard to follow; if a p number is provided the test used should be provided for that p value within the same bracket.  There are three tests used Chi squared, Pearson and ANOVA.  Very hard to understand what variables and treatments are being used unless the test is provided next to each p number; e.g. (p=0.03, Chi-squared) or (p=1, Pearson correlation).  For the latter a Pearson r should also be provided e.g. (p=1, r=0.1, Pearson correlation).  Also very unclear is how many groups are being compared in the ANOVAs; ordinarily if there are two groups and the data is parametric then a t test is used, or a Welch's t-test if variances are not the same.  If non parametric (eg large variance defences, inappropriate skewness or kurtosis), neither ANOVA or t tests can be used.  If a 1 way ANOVA is being used for parametric data comparing two variables then this is essentially a t test.  Or is a 2 way ANOVA being applied (i.e with 2 categorical variable?) if so the two categorical variable need to be described.  For correaltions of non parametric data a Spearman correlation might be used. 

Another major issue is that the Discussion reads like the introduction to a thesis and does not discuss the results from the paper and is overly long.  It is usual to start with the main findings of the paper, then followed by how they relate, or not, to other studies in the field.  The discussion should not include any introduction to the field.  Only those issues that the study has brought to light should be discussed in context with other findings elsewhere.  Finally, some implications for the field , and suggestions to improve clinical management, if appropriate. Limitations section is good.

Other issues

-Materials and Method should start with an Ethics statement.

-Table 1.  The legend is quite uninformative.  Count of what, % or what, n= what, SD of what?  All would be good to explain in the legend.  What is mean and SD missing for most of the data?

-L32 IMV and NIMV need to be written in full in the abstract and explained in the introduction, otherwise have to wait till l106 before we know what this means.  Also perhaps “received” rather than “were given” l162.  L190 does mechanical ventilation refer to both ? If so why no use IMV/NIMV to reduce confusion.

-L91 – usual to add the town as well.

-Fig 1 "did not performed at" - was not performed at. 

-L169 What does environment mean – urban vs rural? Needs to be clarified.

-L174 What does origin refer to? Ethnic origin, country of origin?

-L195 How can IMV or NIMV itself correlate with anything? Absence/present, IMV vs. NIMV, or duration? Catagorial or continous -very unclear.

-L231 Please define metabolic syndrome as used herein, again is this presence absence and thus Chi squred or is this a scale and Pearson?

-Figure. 2 Again the legend is very uninformative – please highlight features of interest with arrows or similar on the images and annotate in the legend.  Is this 3 months post onset? Post diagnosis? Post hospitalisation?

-Is section 3.8 referring to data in a Table 2? It is not referenced in this section.

-Table 2. Legend again should explain what is described here.  7 mg/dL CRP on both days? 7 patients?  

Throughout the words “was correlated” or “were correlated” are used – confusing use of English. Presume the authors mean that there was a significant correlation, i.e. x correlated with y; “was correlated” really refers to the process rather than the result.  Needs rephrasing throughout.; eg ferritin levels correlated significantly with duration of IMV (p=0.034, r=X.X, Pearson Correlation). 

Reviewer 2 Report

This study analyzes the fibrotic evolution of covid-19 cases followed between 2020 and 2021. The authors question whether the subject is still relevant after the end of the pandemic. The answer to this question can be obtained from the new elements that can be obtained from the study and which could be applied in other epidemics. The authors claim to have two goals: i) to analyze and describe the main risk factors associated with the development of post-COVID-19 pulmonary fibrosis and ii) analyze the relationship between mechanical ventilation and the degree of long-term pulmonary lesions.

1.       The literature on these topics is very vast and the authors should tryed to frame their study within it. In reality, the authors compare their observations mainly with articles published during the pandemic, while references to publications from 2023 are almost completely missing. The most recent studies have placed emphasis on molecular mechanisms [Kooistra EJ, Dahm K, van Herwaarden AE, Gerretsen J, Nuesch Germano M, Mauer K, Smeets RL, van der Velde S, van den Berg MJW, van der Hoeven JG, Aschenbrenner AC, Schultze JL, Ulas T, Kox M, Pickkers P. Molecular mechanisms and treatment responses of pulmonary fibrosis in severe COVID-19. Respir Res. 2023 Aug 9;24(1):196. doi: 10.1186/s12931-023-02496-1.] or growth factors [Dülger SU, Mutlu N, Ceylan Ä°, Özhan E. The relationship between lung fibrosis, the epidermal growth factor receptor, and disease outcomes in COVID-19 pneumonia: a postmortem evaluation. Clin Exp Med. 2023 Aug;23(4):1181-1188. doi: 10.1007/s10238-022-00872-7].

2.       The most recent literature on clinical trials includes Multicenter studies [Cilli A, Hanta I, Uzer F, Coskun F, Sevinc C, Deniz PP, Parlak M, Altunok E, Tertemiz KC, Ursavas A. Characteristics and outcomes of COVID-19 patients with IPF: A multi-center retrospective study. Respir Med Res. 2022 May;81:100900. doi: 10.1016/j.resmer.2022.100900] and clinical contributions [Sturgill JL, Mayer KP, Kalema AG, Dave K, Mora S, Kalantar A, Carter DJ, Montgomery-Yates AA, Morris PE. Post-intensive care syndrome and pulmonary fibrosis in patients surviving ARDS-pneumonia of COVID-19 and non-COVID-19 etiologies. Sci Rep. 2023 Apr 21;13(1):6554. doi: 10.1038/s41598-023-32699-x. ] which could be useful to include in the comparison with Romanian observations.

3.       Line 125. The authors indicate the term gender. In the rest of the article, they do not indicate that they have carried out any investigations on gender. It is more appropriate to use the term "sex".

Round 2

Reviewer 2 Report

The authors have deeply revised the manuscript